USP22 as a key regulator of glycolysis pathway in osteosarcoma: insights from bioinformatics and experimental approaches

Zhang Qiao
Zhu Jinwei
Xie Jian
Gu Yurong
http://orcid.org/0009-0005-7057-9526 Chen Lu ndefy06045@ncu.edu.cn
Department of Orthopaedics, The Second Affiliated Hospital of Nanchang University , Nanchang, Jiangxi , China
Zhang Xin
Electronic publication date: 2024 May 20
Publication date: 2024
Volume: 12
Electronic Location ID: e17397
Received 2023 Sep 19; Accepted 2024 Apr 25
Copyright: © 2024 Zhang et al.
Copyright year: 2024
Copyright holder: Zhang et al.
License: This is an open access article distributed under the terms of the Creative Commons Attribution License, which permits unrestricted use, distribution, reproduction and adaptation in any medium and for any purpose provided that it is properly attributed. For attribution, the original author(s), title, publication source (PeerJ) and either DOI or URL of the article must be cited.
License URL: https://creativecommons.org/licenses/by/4.0/

Keywords: Ubiquitin-specific processing peptidase 22 (USP22), Osteosarcoma, Glycolysis, Bioinformatics, Experimental validation

Funding: Key Projects of Science and Technology Plan of Jiangxi Provincial Department of Education GJJ210102 This work was supported by the Key Projects of Science and Technology Plan of Jiangxi Provincial Department of Education: No. GJJ210102. The funders had no role in study design, data collection and analysis, decision to publish, or preparation of the manuscript.

==============================
Background

Osteosarcoma is the most common primary malignant bone tumor, but its pathogenesis remains unclear. Ubiquitin-specific processing peptidase 22 (USP22) is reported to be highly expressed and associated with tumor malignancy and prognosis in cancers. However, the role and mechanism of USP22 in osteosarcoma is not fully understood. This study aims to investigate the function and potential mechanism of USP22 in osteosarcoma using bioinformatics analysis combined with experimental validation.

Methods

We first integrated transcriptomic datasets and clinical information of osteosarcoma from GEO and TCGA databases to assess the expression and prognostic value of USP22 in osteosarcoma. Then, differential expression analysis and weighted gene co-expression network analysis (WGCNA) were conducted to identify USP22-related co-expressed genes. Gene Ontology (GO) and Kyoto Encyclopedia of Genes and Genomes (KEGG) enrichment analyses were performed to explore the biological functions and signaling pathways of USP22 co-expressed genes. To validate the accuracy of bioinformatics analyses, we downregulated USP22 expression in osteosarcoma cell line Sao-2 using siRNA and assessed its effect on cell proliferation, migration, invasion, apoptosis, and regulation of key signaling pathways.

Results

We found that USP22 was highly expressed in osteosarcoma tissues and correlated with poor prognosis in osteosarcoma patients. USP22 also showed potential as a diagnostic marker for osteosarcoma. In addition, 344 USP22-related co-expressed genes were identified, mainly involved in signaling pathways such as glycolysis, oxidative phosphorylation, spliceosome, thermogenesis, and cell cycle. The in vitro experiments confirmed the accuracy and reliability of bioinformatics analyses. We found that downregulation of USP22 could inhibit Sao-2 cell proliferation, migration, invasion, and induce apoptosis. Furthermore, downregulation of USP22 significantly reduced aerobic glycolysis levels in Sao-2 cells and inhibited the expression of key enzymes and transporters in aerobic glycolysis pathways such as HK2, PKM2, and GLUT1.

Conclusions

USP22 plays a critical role in the occurrence, development, and prognosis of osteosarcoma. USP22 could influence Sao-2 cell proliferation, apoptosis, migration, and invasion by regulating the glycolysis pathway, thereby promoting osteosarcoma progression. Therefore, USP22 may be a potential therapeutic target for the treatment of osteosarcoma.

Introduction

Osteosarcoma (OS) is a malignant neoplasm that commonly arises in adolescents and young adults, with the major characteristic that tumor cells are capable of producing bone tissue (Corre et al., 2020). It is a rare cancer, with an incidence rate of approximately 1–3 cases per million people per year, but accounts for around 20% of all bone tumors (Zhao et al., 2021; Kansara et al., 2014). Osteosarcoma can occur in any bone, but the most common site is the metaphysis of long bones such as the femur, tibia, and humerus. It can also occur in other skeletal sites such as the pelvis, ribs, and spine (Corre et al., 2020; Luetke et al., 2014). The main clinical manifestations are bone pain, mass and impaired motor function. Radiographic features include bone destruction, soft tissue mass and periosteal reaction (Vergel De Dios et al., 1992; Muratori et al., 2019). Due to the high malignancy and early metastasis of osteosarcoma, patients generally have very poor prognosis. Studies found the 5-year overall survival rates range from 65–75% for localized OS patients, while only around 20% for recurrent and metastatic tumors (Miwa et al., 2019). Currently, the main therapeutic approaches for osteosarcoma include surgery, chemotherapy and radiotherapy. However, these treatments still have certain limitations, especially for patients with advanced diseases where the efficacy is often unsatisfactory. In addition, osteosarcoma has a high recurrence rate and treatment becomes more difficult after recurrence, leading to even worse prognosis (McGuire et al., 2017). Therefore, it is of great medical and social significance to identify novel therapeutic targets for osteosarcoma.

The pathogenesis of osteosarcoma involves multiple factors, including genetics, environment and lifestyle. Familial inheritance, radiation and chemical exposure, high intensity training, obesity and other factors may increase the risk of osteosarcoma (Simpson & Brown, 2018). The exact pathogenic mechanisms of osteosarcoma remain largely unclear. It is currently believed to be associated with abnormal expression of certain genes, tumor microenvironment changes, glycolysis and other mechanisms (Lilienthal & Herold, 2020; Liu et al., 2023). For example, inactivation of tumor suppressor genes like P53, and overexpression of oncogenes such as c-myc and c-fos can promote the initiation and progression of osteosarcoma (Li et al., 2021; Han et al., 2021; Wang et al., 2021).

USP22 (ubiquitin-specific processing peptidase 22) is a deubiquitinating enzyme and a subunit of the human Spt-Ada-Gcn5-acetyltransferase (hSAGA) complex. Its encoding gene is located at 17p11.2 in the human genome, consisting of 14 exons (Zhang et al., 2008; Ren et al., 2022). USP22 has the biological activities of covalently binding substrate proteins and regulating protein degradation, playing important roles in cell cycle, apoptosis and tumorigenesis (Feng et al., 2021; Guo et al., 2022). Studies show USP22 is abnormally expressed in many tumor diseases and closely associated with cancer initiation, progression and prognosis. For instance, Liu et al. (2019) found USP22 was upregulated in gastric cancer and could promote growth and metastasis via activating c-Myc/NAMPT/SIRT1-dependent FOXO1 and YAP signaling. Kosinsky et al. (2019) showed that downregulation of USP22 can reduce the H3K9ac level on the HSP90AB1 gene, inhibit HSP90AB1 expression, thereby inhibiting the growth of breast cancer cells and colon cancer cells, and increasing the therapeutic sensitivity of the anticancer drug Ganetespib. Therefore, USP22 is a promising therapeutic target worthy of investigation. However, the role and mechanism of USP22 in osteosarcoma remains rarely reported. Here, we will utilize bioinformatics approaches to explore the function of USP22 in osteosarcoma initiation, progression and prognosis, identify the potential mechanisms, and further validate experimentally the effects of USP22 on osteosarcoma cell growth and underlying mechanisms. Our findings will provide new insights and targets for osteosarcoma treatment, and help elucidate the pathogenic mechanisms.

Materials and Methods

Ethical statement

This study obtained ethical support from the Ethics Committee of the Second Affiliated Hospital of Nanchang University (No. 2022021). All procedures performed in studies involving human participants were in accordance with the 1964 Helsinki Declaration and its later amendments or comparable ethical standards.

Transcriptomic datasets acquisition and preprocessing

Two transcriptomic datasets related to osteosarcoma (GSE16088 and GSE21257) were obtained from GEO database (https://www.ncbi.nlm.nih.gov/geoprofiles/). GSE16088 contains expression data of six normal and 14 tumor tissues. GSE21257 contains transcriptomic data and clinical information of 53 osteosarcoma patients. The expression profiles of GSE16088 and GSE21257 were preprocessed including gene symbol conversion, normalization and median polish. Based on The Cancer Genome Atlas (TCGA) database (https://portal.gdc.cancer.gov/), transcriptomic data and clinical information of 88 osteosarcoma patients (TCGA-OS cohort) were acquired. The transcriptomic data was also preprocessed.

Analysis of USP22 expression and prognosis in osteosarcoma

To investigate the potential role of USP22 in osteosarcoma, the expression of USP22 was firstly analyzed in tumor and normal tissues using the GSE16088 dataset. Receiver operating characteristic (ROC) analysis was then performed to assess the diagnostic value of USP22 in differentiating between osteosarcoma and normal tissues. This analysis was carried out using the R package “pROC”.

Subsequently, the relationship between USP22 expression and prognosis in osteosarcoma patients was examined using Kaplan-Meier survival curve analysis based on the GSE21257 dataset and TCGA-OS cohort. To evaluate the sensitivity and specificity of USP22 in predicting the prognosis of osteosarcoma patients, ROC analysis was performed at 1, 3, and 5-year intervals using the R package “survivalROC”.

Identification of USP22 co-expressed genes in osteosarcoma

Firstly, differentially expressed genes (DEGs) between tumor (high USP22) tissues and normal (low USP22) tissues were identified from GSE16088 dataset using limma package in R software. DEGs were filtered with criteria of absolute fold change ≥2 and p < 0.01. These DEGs were correlated with USP22 expression in GSE16088 dataset.

Then, weighted gene co-expression network analysis (WGCNA) was performed on TCGA-OS cohort to identify genes co-expressed with USP22 using R package “WGCNA”. The procedures were: firstly, a soft thresholding power was selected for network construction based on scale-free topology. It transformed the adjacency matrix to continuous values between 0 and 1 for a scale-free network close to biological network state. Secondly, blockwiseModules function was utilized to construct the unsigned co-expression network, and modules containing genes with similar expression patterns were identified using average linkage hierarchical clustering and DynamicTreeCut algorithm. Module eigengenes (MEs) were calculated as the first principal component of modules and used to evaluate association with clinical traits. Modules with p < 0.05 were considered significantly associated with the trait. Finally, gene significance (GS) and module membership (MM) were used as filtering criteria to select hub genes in the USP22 co-expression module, with the thresholds set at GS > 0.2 and MM > 0.6.

The common genes identified from both GSE16088 and TCGA datasets were considered as USP22 co-expressed genes in osteosarcoma.

PPI network analysis and functional enrichment analysis

Protein-protein interaction (PPI) analysis was performed for the USP22 co-expressed genes using STRING database (https://string-db.org/) to explore their potential interactions. The PPI network was visualized using Cytoscape software (version 3.7.2).

Then, to investigate the biological functions and signaling pathways involved with USP22 in osteosarcoma, 344 USP22 co-expressed genes (common genes) were imported into DAVID database (https://david.ncifcrf.gov/) for functional enrichment analysis, including Gene Ontology (GO) and Kyoto Encyclopedia of Genes and Genomes (KEGG) pathway enrichment analysis. Significantly enriched GO terms and KEGG pathways (p < 0.05) were visualized using R software. The most significant KEGG pathway is considered the key signaling pathway associated with USP22 in osteosarcoma.

Gene set enrichment analysis (GSEA) was performed using GSEA software version 4.0.1 to identify differentially expressed signaling pathways between USP22 low expression and high expression groups in GSE16088 dataset and TCGA-OS cohort, with KEGG gene sets as reference. The normalized enrichment score (NES) >0 and p < 0.05 represent significantly upregulated signaling pathways, while NES < 0 and p < 0.05 represent significantly downregulated signaling pathways.

Cell culture and transfection

The human osteosarcoma cell line Sao-2 was obtained from the Cell Bank of the Chinese Academy of Sciences (Shanghai, China) and cultured in DMEM medium (Gibco, CA, USA) supplemented with 10% fetal bovine serum (FBS) (Gibco, Waltham, MA, USA) and 1% penicillin/streptomycin (Solarbio, Beijing, China) at 37 °C in a humidified atmosphere containing 5% CO2.

USP22 expression was silenced by transfecting Sao-2 cells with USP22-specific small interfering RNA (siUSP22) using Lipofectamine 3000 (Invitrogen, Carlsbad, CA, USA) according to the manufacturer’s protocol. The sequences of siRNA were as follows: 5′-GAAGCAUAUUCACGAGCAUTT-3′ (forward) and 5′-AUGCUCGUGAAUAUGCUUCTT-3′ (reverse). Non-targeting siRNA (siNC) was used as control. At 24 h post-transfection, Western blot analysis was conducted to assess USP22 protein levels in normal Sao-2 cells, siNC-transfected Sao-2 cells, and siUSP22-transfected Sao-2 cells to evaluate the knockdown efficiency of siUSP22.

Cell proliferation assay

The effect of USP22 knockdown on cell proliferation was evaluated using the Cell Counting Kit-8 (CCK-8) (NCM Biotech, Suzhou, China) assay. Briefly, transfected cells were seeded into 96-well plates at a density of 2,000 cells per well and incubated for 24, 48, and 72 h. At each time point, 10 µl of CCK-8 solution was added to each well, and the cells were incubated for 30 min at 37 °C. The absorbance at 450 nm was measured using a microplate reader (BioTek, Winooski, VT, USA).

Colony formation assay

Suspended Sao-2 cells were seeded at a density of 500 cells per well in a 6-well plate and cultured for 14 days to allow for colony formation. After 14 days, the cells were then fixed with 4% paraformaldehyde and stained with 0.1% crystal violet. The number of colonies with more than 50 cells was counted under a microscope (Nikon, Tokyo, Japan).

Scratch assay

Transfected cells were seeded into 6-well plates and grown to confluence. A scratch was made in the cell monolayer using a 200-µl pipette tip, and the cells were washed with PBS to remove the debris. The cells were then incubated in serum-free medium for 24 h, and the migrated cells were photographed under a microscope (Nikon, Tokyo, Japan) and quantified using ImageJ software.

Transwell invasion assay

Single-cell suspensions from the USP22 knockdown group and the siNC group were seeded into the upper chamber of Transwell inserts (Corning, NY, USA) coated with Matrigel (BD Biosciences, Franklin Lakes, NJ, USA), while the lower chamber was filled with 10% FBS-containing medium. After 24 h of incubation, the invaded cells were fixed with 4% paraformaldehyde and stained with 0.1% crystal violet. The number of invaded cells was counted under a microscope (Nikon, Tokyo, Japan).

Cell apoptosis assay

Single-cell suspensions were harvested and washed with cold PBS. The cells were then stained with Annexin V-FITC and propidium iodide (PI) using the Annexin V-FITC Apoptosis Detection Kit (Solarbio, Beijing, China) according to the manufacturer’s instructions. The stained cells were analyzed using a flow cytometer (Beckman Coulter, Brea, CA, USA).

Glucose consumption, lactate production, and ATP level assays

Transfected cells were seeded into 12-well plates and cultured for 48 h. The supernatant was collected, and the glucose and lactate levels were measured using the Glucose Assay Kit (Solarbio, Beijing, China) and Lactate Assay Kit (Solarbio, Beijing, China), respectively. The intracellular ATP level was measured using the ATP Assay Kit (Solarbio, Beijing, China) according to the manufacturer’s instructions.

Seahorse analysis of glycolysis

The extracellular acidification rate (ECAR) was measured in siNC and siUSP22 transfected Sao-2 cells using a Seahorse XF96 Extracellular Flux Analyzer (Agilent Technologies, Santa Clara, CA, USA) to assess glycolysis. Briefly, 2 × 104 cells per well were seeded into XF96 cell culture microplates and incubated overnight. Before measurements, cells were washed and incubated in XF base medium containing 1 mM pyruvate, 2 mM glutamine and 10 mM glucose. The ECAR was monitored under basal conditions and in response to the sequential addition of glucose (10 mM), oligomycin (1 μM) and 2-deoxyglucose (50 mM). The glycolysis and glycolytic capacity were calculated based on the ECAR values.

Quantitative real-time PCR

Total RNA was extracted from siNC and siUSP22 transfected Sao-2 cells using TRIzol reagent (Invitrogen, Carlsbad, CA, USA) according to the manufacturer’s protocol. The concentration and purity of extracted RNA were determined using a NanoDrop 2000 spectrophotometer (Thermo Fisher Scientific, Waltham, MA, USA). Complementary DNA was generated from 1 microgram of total RNA through reverse transcription according to the manufacturer’s protocol of a cDNA synthesis kit (Takara, Dalian, China). Polymerase chain reaction was performed on a real-time amplification system (Applied Biosystems, CA, USA) employing a premix reagent (Takara, Dalian, China). The PCR procedure involved initial denaturation at 95 °C for 30 s, followed by 40 cycles consisting of denaturation at 95 °C for 5 s and annealing/extension at 60 °C for 34 s. The primer sequences were:

USP22 forward, 5′-GGAAAATGCAAGGCGTTGGAG-3′, reverse, 5′-GTGCAGTTCGAGGTGATCTTT-3′; HK2 forward, 5′-GAGCCACCACTCACCCTACT-3′, reverse, 5′-CCAGGCATTCGGCAATGTG-3′; PKM2 forward, 5′-ATGTCGAAGCCCCATAGTGAA-3′, reverse, 5′-TGGGTGGTGAATCAATGTCCA-3′; GLUT1 forward, 5′-GGCCAAGAGTGTGCTAAAGAA-3′, reverse, 5′-ACAGCGTTGATGCCAGACAG-3′; β-actin forward, 5′-AGATTACTGCCCTGGCTCCTAG-3′, reverse, 5′-CATCGTACTCCTGCTTGCTGAT-3′. The relative expression levels were calculated using the 2−ΔΔCt method. β-actin was used as an internal control.

Western blot analysis

The expression levels of USP22, HK2, PKM2, and GLUT1 in cells were evaluated using Western blot analysis. Briefly, transfected cells were harvested and lysed in RIPA buffer (NCM Biotech, Suzhou, China) containing protease inhibitor cocktail (NCM Biotech, Suzhou, China). The protein concentration was determined using the BCA Protein Assay Kit (NCM Biotech, Suzhou, China). Equal amounts of protein were separated by SDS-PAGE and transferred to PVDF membranes (Millipore, Bedford, MA, USA). The membranes were then blocked with 5% non-fat milk in TBS-T buffer and incubated with primary antibodies against USP22, HK2, PKM2, GLUT1, and β-actin (Abcam, Cambridge, UK) at 4 °C overnight. After washing with TBS-T buffer, the membranes were incubated with HRP-conjugated secondary antibodies (Abcam, Cambridge, UK) for 1 h at room temperature. The protein bands were visualized using ECL reagent (NCM Biotech, Suzhou, China) and quantified using ImageJ software.

Statistical analysis

All experimental data were shown from at least three independent experiments and presented as the mean ± standard deviation (SD). Differences between groups were analyzed using the Student’s t-test in GraphPad Prism 7.0. A p value less than 0.05 was considered statistically significant.

Results

USP22 is highly expressed in osteosarcoma and associated with poor prognosis

In GSE16088 dataset, USP22 expression was significantly higher in tumor tissues than normal tissues (p < 0.05) (Fig. 1A). ROC curve analysis showed the area under curve (AUC) of USP22 was 1.00, indicating its high diagnostic value (Fig. 1B).

Figure 1 Expression and prognostic analysis of USP22 in osteosarcoma.

(A) Expression of USP22 in osteosarcoma and normal tissues from the GSE16088 dataset. (B) ROC analysis of USP22 expression in the GSE16088 dataset for diagnosing osteosarcoma. (C) Kaplan-Meier survival analysis of USP22 in the GSE21257 dataset. (D) ROC analysis of USP22 in the GSE21257 dataset for predicting 3-, 5-, and 10-year survival. (E) Kaplan-Meier survival analysis of USP22 in the TCGA-OS cohort. (F) ROC analysis of USP22 in the TCGA-OS cohort for predicting 3-, 5-, and 10-year survival.

Kaplan-Meier survival analysis using the GSE21257 dataset revealed that high USP22 expression was associated with poor prognosis of osteosarcoma patients (p < 0.05) (Fig. 1C). ROC analysis showed the AUC of USP22 for predicting 3, 5 and 10-year survival were 0.79, 0.77 and 0.76, respectively (Fig. 1D). These results suggest that USP22 has good prognostic value in osteosarcoma. Similar results were obtained using the TCGA dataset. Kaplan-Meier survival analysis showed that high USP22 expression correlated with poor prognosis of osteosarcoma patients (p < 0.05) (Fig. 1E). The AUC of USP22 for predicting 3, 5 and 10-year survival were 0.74, 0.79 and 0.81, respectively (Fig. 1F). These results suggest that USP22 has good prognostic value in osteosarcoma.

Identification of USP22 co-expressed genes

A total of 4,536 DEGs were identified between tumor (high USP22) and normal (low USP22) groups from the GSE16088 dataset, including 2,386 upregulated and 2,150 downregulated DEGs (Figs. 2A and 2B).

Figure 2 Identification of USP22-related differentially expressed genes (DEGs) in the GSE16088 dataset.

(A) Volcano plot of differential expression analysis between tumor tissues (USP22 high expression group) and normal tissues (USP22 low expression group). (B) Heatmap of DEGs expression in tumor tissues (USP22 high expression group) and normal tissues (USP22 low expression group).

WGCNA was performed on TCGA dataset to identify USP22 co-expressed genes. The power of β = 5 was selected as the optimal soft thresholding to ensure a scale-free network (Fig. 3A). Average linkage hierarchical clustering classified the genes into 24 modules (Fig. 3B). The connectivity analysis showed the distances between modules were greater than 0.25, indicating good independence (Fig. 3C). Among the modules, blue, turquoise, lightgreen, magenta and lightyellow were significantly correlated with USP22 expression (p < 0.05) (Fig. 3D). A total of 1,345 hub genes were identified from these five modules based on criteria of GS >0.2 and MM >0.6 (Figs. 4A–4E).

Figure 3 Identification of USP22-related co-expressed genes in the TCGA-OS dataset using weighted gene co-expression network analysis (WGCNA).

(A) Determination of the optimal soft-thresholding power (β). (B) Hierarchical clustering dendrogram of genes. The genes were classified into 24 modules indicated by different colors. (C) Heatmap of the module-trait relationships. The distances between modules were greater than 0.25, indicating good independence. (E) Module-trait relationships between USP22 expression and gene modules.

Figure 4 Identification of key genes co-expressed with USP22.

(A–E) Hub genes were identified from the blue (A), turquoise (B), lightgreen (C), magenta (D), and lightyellow (E) modules based on criteria of GS > 0.2 and MM > 0.6.

By comparing the DEGs from TCGA-OS cohort and hub genes from GSE16088 dataset, 344 common genes were obtained (Fig. 5). The genes co-expressed with USP22 may have a close regulatory relationship with USP22 in the occurrence, development, and prognosis of osteosarcoma.

Figure 5 Common genes identified by comparing the DEGs from TCGA-OS cohort and hub gens from GSE16088 dataset.

PPI network analysis and functional enrichment analysis

The PPI network analysis revealed USP22 interacted directly with several co-expressed genes including ENY2, TAF12, RPS27A, MYC, PSMD14, RSL24D1 (Fig. 6). This implies USP22 may participate in the development and prognosis of osteosarcoma through interactions with these co-expressed genes.

Figure 6 Protein-protein interaction (PPI) network analysis.

PPI network analysis showing the interactions between USP22 and its co-expressed genes. Blue nodes represent the direct interacting partners of USP22, while green nodes represent indirect interactors in the network.

Enrichment analysis was performed on the 344 genes co-expressed with USP22 using KEGG and GO databases. GO analysis includes three categories: biological processes (BPs), cellular components (CCs), and molecular functions (MFs). Figure 7A displays the top 10 significantly enriched BPs, CCs, and MFs ranked by count. The result shown that these 344 co-expressed genes are mainly involved in 92 biological processes, such as translation, cytoplasmic translation, mRNA splicing via spliceosome, and proteasome-mediated ubiquitin-dependent protein catabolic processes. Additionally, they were found to be associated with 77 cellular components, such as cytosol, nucleus, cytoplasm, and nucleoplasm, and 49 molecular functions, such as protein binding, RNA binding, identical protein binding, and structural constituent of ribosome. The KEGG enrichment analysis revealed that these 344 co-expressed genes were mainly enriched in 17 signaling pathways, such as metabolic pathways, glycolysis/gluconeogenesis, oxidative phosphorylation, spliceosome, thermogenesis, and cell cycle (Fig. 7B). Most of these pathways were related to metabolism, with glycolysis/gluconeogenesis being one of the most significantly enriched pathways among them (Fig. 7C). As shown in Fig. 7D, the metabolism-related pathways and their associated genes are mapped. GSEA showed glycolysis/gluconeogenesis pathway was significantly upregulated in USP22 high expression group compared to low expression group (Figs. 7E and 7F). Therefore, glycolysis/gluconeogenesis was considered a key signaling pathway associated with USP22 in osteosarcoma.

Figure 7 Functional enrichment analysis of 344 co-expressed genes.

(A) Functional enrichment analysis of the 344 co-expressed genes using Gene Ontology (GO) databases, displaying the top 10 significantly enriched biological processes (BPs), cellular components (CCs), and molecular functions (MFs) ranked by count. (B) Kyoto Encyclopedia of Genes and Genomes (KEGG) enrichment analysis of the 344 co-expressed genes, showing the 17 significantly enriched signaling pathways. (C) Classification of KEGG signaling pathways. (D) A map of the metabolism-related pathways and their associated genes identified in KEGG enrichment analysis. (E) Gene Set Enrichment Analysis (GSEA) analysis of glycolysis/gluconeogenesis pathway in the GSE16088 dataset. (F) GSEA analysis of glycolysis/gluconeogenesis pathway in the TCGA-OS cohort.

Downregulation of USP22 inhibits Sao-2 proliferation

Small interfering RNA against USP22 (siUSP22) was used to knockdown USP22 expression in Sao-2 cells. Real-time PCR and western blot analyses showed that siUSP22 exhibited effective silencing efficiency (p < 0.05) (Figs. 8A–8C). CCK-8 and colony formation assays were performed to evaluate cell viability. The results showed that cell viability and growth rates were significantly reduced in the USP22 knockdown group (siUSP22) compared to the control group (siNC) at 24, 48, and 72 h timepoints (p < 0.05). The inhibitory effect of USP22 knockdown on Sao-2 cell growth became more pronounced with longer incubation times (Fig. 8D). Colony formation assay findings indicated that USP22 expression knockdown significantly impaired the colony formation ability of Sao-2 cells compared to the control group (siNC) (p < 0.05) (Figs. 8E and 8F). These results demonstrate that inhibition of USP22 expression can suppress Sao-2 cell proliferation.

Figure 8 The effect of USP22 knockdown on osteosarcoma cell proliferation.

(A) Real-time PCR analysis of USP22 mRNA levels in normal Sao-2 cells (NC) and Sao-2 cells transfected with siRNA against USP22 (siUSP22) and non-targeting control siRNA (siNC). (B) Western blot analysis of USP22 protein expression in the NC, siNC and siUSP22 groups. (C) Quantitative analysis of USP22 protein levels in the NC, siNC and siUSP22 groups. (D) Cell Counting Kit-8 (CCK-8) assay results of Sao-2 cells transfected with siUSP22 or siNC at 24, 48, and 72-h time points. (E) Representative images of colony formation assay of Sao-2 cells transfected with siUSP22 or siNC. (F) Quantitative analysis of colony formation assay showing the colony formation ability of Sao-2 cells in the siNC and siUSP22 group. The values shown in the graph represent the mean ± SD for triplicate samples, as determined by the Student’s t-test. **p < 0.01.

Downregulation of USP22 inhibits migration and invasion of Sao-2 cells

To assess the effect of USP22 expression knockdown on Sao-2 cell migration and invasion, we performed migration and Transwell invasion assays. The migration assay results showed that the migratory ability of Sao-2 cells with USP22 knockdown was significantly decreased compared to the control group, as determined by the wound closure width (p < 0.05). The wound healing rates at 24 h for the control group and USP22 knockdown group were 72.92% and 45.79%, respectively (Figs. 9A and 9B). In the Transwell invasion assay, the number of Sao-2 cells that penetrated the membrane was significantly lower in the USP22 knockdown group compared to the control group (p < 0.05). Compared to the control, the number of Sao-2 cells that penetrated into the lower chamber decreased by 51.48% with USP22 knockdown (Figs. 9C and 9D). These results indicate that inhibition of USP22 expression can suppress Sao-2 cell migration and invasion.

Figure 9 The effect of USP22 downregulation on migration and invasion of osteosarcoma cells.

(A) Representative images of the migration assay of Sao-2 cells transfected with siUSP22 or siNC. (B) Quantitative analysis of the migration assay showing the area of wound recovery after 24 h. (C) Representative images of the Transwell invasion assay of Sao-2 cells in siNC and siUSP22 group. (D) Quantitative analysis of the Transwell invasion assay showing the number of Sao-2 cells that penetrated into the lower chamber. The values shown in the graph represent the mean ± SD for triplicate samples, as determined by the Student’s t-test. **p < 0.01.

Downregulation of USP22 induces apoptosis in Sao-2 cells

The effect of USP22 knockdown on cell apoptosis was evaluated using flow cytometry. The results showed that compared with the siNC group, the apoptosis rate of the siUSP22 group significantly increased from 1.93% to 25.2% (p < 0.05) (Figs. 10A and 10B). The results demonstrate that inhibition of USP22 expression can induce apoptosis in Sao-2 cells.

Figure 10 The effect of USP22 downregulation on apoptosis in osteosarcoma cells.

(A) Representative flow cytometry plots of apoptosis analysis of osteosarcoma cells in siNC and siUSP22 groups. (B) Quantitative analysis of apoptosis rate in siNC and siUSP22 groups. The values shown in panel B represent the mean ± SD for triplicate samples, as determined by the Student’s t-test. **p < 0.01.

Downregulation of USP22 inhibits aerobic glycolysis levels in Sao-2 cells

Enhanced aerobic glycolysis capacity is an important mechanism for rapid tumor cell growth. Inhibiting aerobic glycolysis levels can suppress Sao-2 cell growth. The effect of USP22 knockdown on glycolysis was evaluated by measuring glucose consumption, lactate production, and ATP levels in cells using colorimetric assays. The results showed that compared to the control group, USP22 expression knockdown significantly decreased glucose consumption, lactate production, and ATP generation in Sao-2 cells (p < 0.05). After USP22 knockdown, glucose consumption, lactate production, and ATP generation were reduced to 52.96%, 46.88%, and 46.05% of the control group, respectively (Figs. 11A–11C). Additionally, Seahorse analysis showed that after the addition of glucose, oligomycin and 2-DG, the ECAR of siUSP22 cells was markedly lower than that of siNC control cells, indicating reduced glycolysis capacity (Fig. 11D). Quantitative analysis revealed that glycolysis and glycolytic capacity were significantly decreased in siUSP22 cells compared to siNC cells (p < 0.05) (Fig. 11E).

Figure 11 The effect of USP22 knockdown on aerobic glycolysis levels in osteosarcoma cells.

(A–C) Quantitative analysis of glucose consumption (A), lactate production (B), and ATP levels (C) in the siNC and siUSP22 groups. (D) ECAR curve measured in siNC and siUSP22 Sao-2 cells. (E) Quantification of glycolysis and glycolytic capacity. (F) Real-time PCR analysis of mRNA levels of key glycolytic genes HK2, PKM2 and GLUT1 in siNC and siUSP22 cells. (G) Western blot analysis of glycolysis pathway-related proteins HK2, PKM2, and GLUT1 in siNC and siUSP22 cells. (H) Quantitative analysis of HK2, PKM2, and GLUT1 protein levels in the siNC and siUSP22 groups. The values shown in the graph represent the mean ± SD for triplicate samples, as determined by the Student’s t-test. *p < 0.05, **p < 0.01.

We also detected the mRNA and protein expression changes of key glycolytic enzymes HK2, PKM2 and GLUT1. Real-time PCR results showed that compared with siNC group, the mRNA levels of HK2, PKM2 and GLUT1 were significantly decreased in siUSP22 transfected Sao-2 cells (p < 0.05), with a reduction of 76%, 60% and 51%, respectively (Fig. 11F). Western blot analysis showed that in alignment with mRNA results, the protein levels of HK2, PKM2 and GLUT1 were markedly reduced in siUSP22 group compared to siNC control group (p < 0.05) (Figs. 11G and 11H). These results indicate that inhibition of USP22 expression suppresses Sao-2 cell glycolysis by downregulating both the transcription and translation of glycolytic genes.

Discussion

USP22 has been reported to play an oncogenic role and indicate poor prognosis in various human cancers including gastric, breast and colorectal cancers. It is considered a promising therapeutic target against cancers. Efforts have been made to develop and improve USP22 inhibitors to treat cancers by inhibiting its functions (Melo-Cardenas et al., 2016). The pro-cancer mechanisms of USP22 involve multiple aspects such as glycolysis (Ling et al., 2020), immune microenvironment (Li et al., 2020), lipid metabolism (Ning et al., 2022), oxidative phosphorylation (Zhang et al., 2019), and angiogenesis (Zhang et al., 2019). For example, Ning et al. (2022) found that USP22 is a key factor in fatty acid synthesis, and it can stabilize peroxisome proliferator-activated receptor γ (PPARγ) through deubiquitination, thereby upregulating the expression of acetyl-CoA carboxylase (ACC) and ATP citrate lyase (ACLY) to promote hepatocellular carcinoma growth. Zhang et al. (2019) showed USP22 could facilitate angiogenesis, growth and metastasis of non-small cell lung cancer in vivo. However, few studies demonstrate the inhibitory effects of USP22 on osteosarcoma cells and the specific mechanisms remain unclear.

Integrated analysis of expression profiles using bioinformatics approaches is one of the most effective methods to uncover disease mechanisms, biomarkers and prognostic characteristics with high accuracy and efficiency (Uesaka et al., 2022). In this study, we utilized multiple bioinformatics tools combined with in vitro experiments to investigate the role and mechanisms of USP22 in osteosarcoma. Firstly, by comparing USP22 expression between osteosarcoma and adjacent tissues, we found USP22 was markedly upregulated in osteosarcoma, consistent with previous reports (Zhang et al., 2017). Moreover, survival analysis combined with clinical data revealed, for the first time, high USP22 expression was closely associated with poor prognosis of osteosarcoma patients. Through screening for co-expressed genes with USP22 and functional enrichment analysis, we found that the pro-osteosarcoma mechanisms of USP22 may involve the glycolysis/gluconeogenesis pathway. Finally, in vitro experiments validated the accuracy and reliability of the bioinformatics analysis results. We found that USP22 knockdown could inhibit the glycolytic pathway in osteosarcoma cells, suppress cell proliferation, migration and invasion, and promote cancer cell apoptosis.

Glycolysis is one of the major pathways for cells to generate energy by converting glucose into lactate and producing ATP (Lunt & Vander Heiden, 2011). Under normal conditions, cells utilize both glycolysis and oxidative phosphorylation to produce energy for vital activities. However, in tumor diseases, the role of glycolysis is further enhanced, leading to a series of detrimental effects. The enhanced glycolysis in tumors is mainly due to metabolic remodeling of cancer cells. umor cells tend to be highly dependent on glycolysis and promote glycolysis through various mechanisms to meet demands for energy and biosynthesis, a phenomenon termed “Warburg effect” (Ganapathy-Kanniappan & Geschwind, 2013; Pavlova & Thompson, 2016). Studies show enhanced glycolysis can promote tumor cell growth, metastasis, and resistance to radiotherapy and chemotherapy (Ganapathy-Kanniappan & Geschwind, 2013; Park, Pyun & Park, 2020). Like other tumors, osteosarcoma is also a highly glycolytic tumor, and its initiation, progression and poor prognosis are closely associated with activation of glycolysis (Feng, Ou & Hao, 2022). Therefore, inhibition of glycolysis is an important approach to overcome osteosarcoma growth and metastasis, providing new insights for osteosarcoma treatment. Currently, many anticancer drugs or genes have been found to improve tumor progression in osteosarcoma and other cancers by reversing the “Warburg effect” and suppressing glycolysis. For example, Caudatin, a naturally occurring steroidal glycoside with anticancer activities, was found to inhibit osteosarcoma cell proliferation and invasion by suppressing glycolysis (Zhang et al., 2022). Deng et al. (2022) showed Ubiquitin-like protein FAT10 could promote osteosarcoma glycolysis and growth. Notably, USP22 has been reported to positively regulate glycolysis in a variety of tumors, such as hepatocellular carcinoma and breast cancer (Ling et al., 2020; Li, Gao & Zhang, 2023). Here, we first found that downregulation of USP22 significantly reduced aerobic glycolysis levels in Sao-2 cells and directly suppressed mRNA and protein levels of key glycolytic enzymes HK2 and PKM2, as well as glucose transporter GLUT1. In addition, we found that the molecular mechanism by which USP22 affects the occurrence, development, and prognosis of osteosarcoma may involve multiple signaling pathways such as oxidative phosphorylation, Spliceosome, Thermogenesis, and Cell cycle. In addition, we found that the molecular mechanism by which USP22 affects the occurrence, development and prognosis of osteosarcoma may involve signaling pathways such as oxidative phosphorylation, Spliceosome, Thermogenesis, and Cell cycle, as well as the inter-regulation of USP22 with a variety of co-expressed genes, but this needs to be verified by more in vitro and in vivo experiments.

Conclusion

In summary, our study demonstrates the oncogenic role of USP22 in osteosarcoma. Based on bioinformatics analysis, USP22 is highly expressed in osteosarcoma and associated with poor survival of patients. The molecular mechanisms by which USP22 promotes osteosarcoma involve multiple signaling pathways, including glycolysis, oxidative phosphorylation, Spliceosome, Thermogenesis, and Cell cycle. In vitro experiments further validated the accuracy of our bioinformatics analysis results. Downregulation of USP22 can inhibit the glycolysis and growth of osteosarcoma cells.

Supplemental Information

Supplemental Information 1 Common genes.

Comparison of the DEGs from TCGA-OS cohort and hub genes from GSE16088 dataset, 344 common genes were obtained.

Supplemental Information 2 Functional enrichment analysis.

Functional enrichment analysis of 344 common genes.

Supplemental Information 3 Cell proliferation: CCK8.

Supplemental Information 4 Colony formation.

Supplemental Information 5 ATP content detection.

Supplemental Information 6 Glucose consumption.

Supplemental Information 7 Lactate production.

Supplemental Information 8 Scratch analysis.

Supplemental Information 9 Transwell counting analysis.

Supplemental Information 10 Western blot analysis.

Supplemental Information 11 Western blot analysis of GLUT1.

Supplemental Information 12 Western blot analysis of HK2.

Supplemental Information 13 Western blot analysis of PKM2.

Supplemental Information 14 Western blot analysis of USP22.

Supplemental Information 15 Data of Glycolysis Seahorse Experiment.

Supplemental Information 16 Full-length uncropped blots of Figure 8B.

Supplemental Information 17 Full-length uncropped blots of Figure 11G.

Supplemental Information 18 PCR supplementary experiment results.

Supplemental Information 19 WB supplementary experiment USP22 (uncropped).

Supplemental Information 20 Data of WB supplementary experiment.

Supplemental Information 21 Raw flow cytometry data.

Supplemental Information 22 siNC-1 MFI histograms.

Instrument parameters, gating parameters, and MFI histograms for FACS

Supplemental Information 23 siNC-2 MFI histograms.

Instrument parameters, gating parameters, and MFI histograms for FACS

Supplemental Information 24 siNC-3 MFI histograms.

Instrument parameters, gating parameters, and MFI histograms for FACS

Supplemental Information 25 siUSP22-1 MFI histograms.

Instrument parameters, gating parameters, and MFI histograms for FACS

Supplemental Information 26 siUSP22-2 MFI histograms.

Instrument parameters, gating parameters, and MFI histograms for FACS

Supplemental Information 27 siUSP22-3 MFI histograms.

Instrument parameters, gating parameters, and MFI histograms for FACS

Additional Information and Declarations

Competing Interests

Author Contributions

Human Ethics

DNA Deposition

Data Availability

The authors declare no conflict of interest.

Qiao Zhang conceived and designed the experiments, performed the experiments, analyzed the data, prepared figures and/or tables, authored or reviewed drafts of the article, and approved the final draft.

Jinwei Zhu performed the experiments, analyzed the data, authored or reviewed drafts of the article, and approved the final draft.

Jian Xie performed the experiments, analyzed the data, prepared figures and/or tables, and approved the final draft.

Yurong Gu performed the experiments, prepared figures and/or tables, and approved the final draft.

Lu Chen conceived and designed the experiments, performed the experiments, analyzed the data, prepared figures and/or tables, authored or reviewed drafts of the article, and approved the final draft.

The following information was supplied relating to ethical approvals (i.e., approving body and any reference numbers):

The Second Affiliated Hospital of Nanchang University Medical Research Ethics Committee granted Ethical approval to carry out the study within its facilities.

The following information was supplied regarding the deposition of DNA sequences:

The human osteosarcoma cell line Sao-2 was obtained from the Cell Bank of the Chinese Academy of Sciences (Shanghai, China).

The following information was supplied regarding data availability:

The flow cytometry data is available at Figshare: Chen, Lu (2024). USP22 as a Key Regulator of Glycolysis Pathway in Osteosarcoma: Insights from Bioinformatics and Experimental Approaches. figshare. Dataset. https://doi.org/10.6084/m9.figshare.25450996.v1.

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
