# Peer review of "USP22 as a key regulator of glycolysis pathway in osteosarcoma: insights from bioinformatics and experimental approaches"

_PeerJ, doi:10.7717/peerj.17397_

## Round 0.1 · original submission · Major Revisions

Both reviewers give suggestions for modification. Please revise carefully and answer the reviewer's comments.

Reviewer 1 ·

Basic reporting

no comment

Experimental design

no comments

Validity of the findings

no comment

Additional comments

The authors demonstrated that USP22 was highly expressed in osteosarcoma tissues and correlated with
30 poor prognosis in osteosarcoma patients. Moreover, the results showed that USP22 influenced osteosarcoma cell proliferation, apoptosis, migration, and invasion. The biological function of USP22 may be related with its role in modulation of glycolysis pathway, thereby promoting osteosarcoma progression. However, there are still some points need to be assessed.
1. Figure 6. The dimention in some panels in Figure 6 are unclear. It need to enlarge the characters in it.
2. Figure 10D, concerning western bloting data in Fig 10D, there is no evidence to show the USP22 depletion. It should be included in the results.
3. As for the relationship between USP22 and HK2, PKM2, and GLU1, How does USP22 regulate these proteins? The regulation of USP22 on these proteins is protein level or transcription level in a direct or indirect manner? It should at least mentioned in discussion section.

Reviewer 2 ·

Basic reporting

This is a work describing the role of USP22 in osteosarcoma (OS) progression. Both bioinformatics and experimental approaches were performed. Part of conclusions were similar to a previous work (PMID:27983930), which may affect the novelty of the present work.

Experimental design

Authors focused on the regulation of USP22 on the glycolysis of OS cells. More experiments should be performed, like Seahorse and Transcriptome Sequencing in OS cell lines to further demonstrate the relationship between USP22 and glycolysis in OS cells. Over-expression of USP22 in OS cells and corresponding functional assays should be added.

Validity of the findings

no comment

Additional comments

The name of OS cell line should be "Saos-2".

---

## Round 0.2 · Minor Revisions

All micrographs should be marked with scale.

When you resubmit, please provide raw flow cytometry data in CSV format, together with precise information on instrument parameters, gating parameters, and mean fluorescent intensity (MFI) histograms for FACS.

Reviewer 2 ·

Basic reporting

The authors answered our questions well and added the necessary experiments. It is acceptable.

Experimental design

no comment

Validity of the findings

no comment

Additional comments

The authors answered our questions well and added the necessary experiments.

Reviewer 3 ·

Basic reporting

The authors addressed my concerns and have improved their manuscript. I would recommend "Accept" this time.

Experimental design

No comment

Validity of the findings

No comment

---

## Round 0.3 · accepted · Accept

After revisions, two reviewers agreed to publish the manuscript. I also reviewed the manuscript and found no obvious risks to publication. Therefore, I also approved the publication of this manuscript.